# Targeting NMDA Receptors at the Neurovascular Unit: Past and Future Treatments for Central Nervous System Diseases

**DOI:** 10.3390/ijms231810336

**Published:** 2022-09-07

**Authors:** Célia Seillier, Flavie Lesept, Olivier Toutirais, Fanny Potzeha, Manuel Blanc, Denis Vivien

**Affiliations:** 1Normandie University, UNICAEN, INSERM, GIP Cyceron, Institute Blood and Brain @Caen-Normandie (BB@C), UMR-S U1237, Physiopathology and Imaging of Neurological Disorders (PhIND), 14000 Caen, France; 2Lys Therapeutics, Cyceron, Boulevard Henri Becquerel, 14000 Caen, France; 3Department of Immunology and Histocompatibility (HLA), Caen University Hospital, CHU, 14000 Caen, France; 4Department of Clinical Research, Caen University Hospital, CHU, 14000 Caen, France

**Keywords:** NMDA receptors, central nervous system, inflammation, excitotoxicity, blood-brain barrier, neurological diseases, therapeutic strategies

## Abstract

The excitatory neurotransmission of the central nervous system (CNS) mainly involves glutamate and its receptors, especially N-methyl-D-Aspartate receptors (NMDARs). These receptors have been extensively described on neurons and, more recently, also on other cell types. Nowadays, the study of their differential expression and function is taking a growing place in preclinical and clinical research. The diversity of NMDAR subtypes and their signaling pathways give rise to pleiotropic functions such as brain development, neuronal plasticity, maturation along with excitotoxicity, blood-brain barrier integrity, and inflammation. NMDARs have thus emerged as key targets for the treatment of neurological disorders. By their large extracellular regions and complex intracellular structures, NMDARs are modulated by a variety of endogenous and pharmacological compounds. Here, we will present an overview of NMDAR functions on neurons and other important cell types involved in the pathophysiology of neurodegenerative, neurovascular, mental, autoimmune, and neurodevelopmental diseases. We will then discuss past and future development of NMDAR targeting drugs, including innovative and promising new approaches.

## 1. Introduction

NMDARs are heterotetramers formed by the association of two required GluN1 subunits and either two GluN2 or GluN3 subunits. While the GluN1 subunit is encoded by a single gene, there are four different genes for GluN2 (*GRIN2A-GRIN2D* encoding for GluN2A-GluN2D) and two different genes (*GRIN3A-3B* encoding for GluN3A-3B) conferring to the receptors different biophysical and pharmacological properties as well as distinct expression and signaling profiles [1,2,3]. At least seven splice variants of GluN1 exist, giving also a spectrum of pharmacological properties and interactions with intracellular and extracellular proteins to NMDARs [4]. The location of neuronal NMDARs at the synapse is an important feature of their functions. NMDARs are not restrained to a neuronal expression but have been described in many other cell types, including endothelial, glial, and immune cells [5,6]. In this review, we will focus on neurodegenerative, neurovascular, autoimmune, and mental disorders where cellular mechanisms involving NMDARs are disturbed [7]. The molecules which are known to modulate NMDARs include high and low-affinity channel blockers, competitive antagonists at the L-glutamate or glycine binding sites, selective antagonists along with positive or negative allosteric modulators (PAMs and NAMs, respectively) (Table 1).

### 1.1. Neuronal NMDAR Structure, Distribution, and Functions

#### 1.1.1. NMDARs on Neurons

In the adult forebrain, the most widespread subtypes are the GluN1/GluN2A and GluN1/GluN2B heterodimers, together with the GluN1/GluN2A/GluN2B heterotrimers [20,47] (Table 2). The expression of NMDAR subtypes evolves during development and is highly variable at the level of brain regions, cell types, and subcellular compartments (synaptic vs. extrasynaptic) [20]. Activation of the NMDARs requires the binding of two molecules of agonist (glutamate or an amino acid analog: L-theanine [18]) on the GluN2 or GluN3 subunits, two molecules of co-agonist (glycine or D-serine) on GluN1 subunits and the removal of the Mg^2+^ channel blocker [16,17,48,49,50]. Once activated, NMDARs mediate Ca^2+^ entry through the ion channel. Interestingly, the site of action of neurosteroids is located at the extracellular vestibule of the receptor’s ion channel pore and is accessible after receptor activation [8]. Polyamines, Zn^2+^, and Cu^2+^ ions have been reported as endogenous antagonists of NMDARs [19,20]. Depending on their precise localization at the synapse, i.e., either synaptic or extrasynaptic compartment, NMDARs can mediate opposite effects [51]. Although the main effects of NMDARs are driven by Ca^2+^ influx, activation of NMDARs can also induce non-ionotropic signaling. These Ca^2+^ flow-independent effects, also called metabotropic functions of NMDARs, are an active area of research with contradictory findings and are mediated by scaffolding proteins and intracellular signaling molecules that bind to the C-terminal domain of the GluN2 subunit [52]. The metabotropic NMDAR pathways are also involved in structural plasticity along with long-term depression (LTD) mechanisms [52,53]. The cerebral distribution of NMDAR subtypes is variable according to their structure. GluN2A subunit is present throughout the adult brain, while GluN2B is preferentially expressed in the forebrain. GluN2C subunit is discriminatorily localized in the cerebellum and the olfactory bulb, while the GluN2D subunit is weakly expressed in the cortex, the hippocampus, and the cerebellum [20,54,55]. Spatial subdivision of NMDAR subtypes also exists at the synapse level. They are differently distributed at the pre- and postsynaptic levels. Presynaptic NMDARs have been identified in several types of synapses across the central nervous system (CNS) [56]. The modulation of neurotransmitter release by presynaptic NMDARs plays a role in synaptic plasticity processes [57,58,59]. Postsynaptic NMDARs are named depending on their synaptic distribution. Synaptic NMDARs are conventionally defined as functional receptors which are activated by glutamate release during low-frequency synaptic events [51]. Extrasynaptic NMDARs are not or very little activated by the synaptic release of glutamate but can be activated by glutamate from other sources, such as glial cells or glutamate spillover [51,60]. They represent 2/3 of NMDARs in the early stages of development and 1/3 in adults [61,62].

The synaptic NMDAR distribution is not fixed and is complexified by their lateral mobility [61,63]. NMDARs can move along the plasma membrane between the synaptic and extrasynaptic compartments depending on the levels of receptor activation and phosphorylation [64]. It is interesting to note that this diffusion can be modulated by extracellular factors such as matrix metalloproteases (MMP), tissue-type plasminogen activators (tPA), or co-agonists [46,65,66,67].

Recently, NMDARs have also been identified on interneurons controlling cellular excitation in a synapse-type specific manner, leading to divergent dendritic integration properties [68]. Furthermore, NMDARs have been identified on parvalbumin-expressing (PV^+^) GABAergic interneurons mediating feedback inhibition and as having a key role in gamma oscillations and information processing. Moreover, these receptors enable cooperative recruitment of PV^+^ interneurons, strengthening and stabilizing principal cell assemblies [69]. 

#### 1.1.2. NMDARs in Brain Development

Mammalian cortex development proceeds through a sequence of time-controlled cellular proliferation, differentiation, and migration events. The neocortex is radially organized into six layers, which each are enriched in specialized subtypes of neurons and tangentially organized into areas specialized in distinct functions. Excitatory neurons are generated directly from self-renewing radial glial cells and indirectly from the intermediate progenitors in the intermediate zone, both expressing NMDARs [77]. Several studies suggest that NMDARs are not necessary for the expression of the glutamate AMPARs (α-amino-3-hydroxy-5-methyl-4-isoxazolepropionic acid receptor) in immature synapses [78,79], but rather during the stages of maturation of neural circuits, dendritic maturation and proper synapse refinement, presumably by controlling intrinsic excitability [80,81,82]. The maturation of glutamatergic synapses involves a reorganization of ionotropic glutamate receptors at the pre- and postsynaptic compartment [56]. NMDARs are part of a series of well-controlled cell specification and migration events directly implicated in the brain and neuronal circuits formation and in neurodevelopmental disorders, including cortical malformation, autism, schizophrenia (SCZ), or intellectual disability [83,84,85,86].

#### 1.1.3. NMDARs in LTP and LTD

Long-term potentiation (LTP) and LTD are processed involving persistent strengthening of synapses that leads to a long-lasting increase or a weakening in signal transmission between neurons, respectively. The LTP, along with LTD, is now identified as the molecular support of learning and memory and is part of synaptic plasticity mechanisms [87]. These two forms of synaptic plasticity involve alterations in the excitatory postsynaptic current (EPSC) that is mediated via the activation of AMPARs and NMDARs [88]. The NMDAR-mediated LTP was first described in 1983 by Collingridge and collaborators [89]. NMDAR-dependent LTP and LTD are induced by high-frequency stimulations (few trains of 100 Hz stimulation) and low-frequency stimulations (700–900 trains of 1 Hz stimulation), respectively. It was demonstrated that low stimulations of Schaffer collaterals implicate a non-NMDAR circuit, while intense stimulations involve NMDARs and induce an LTP. NMDARs can be defined as LTP and LTD triggers by their induction roles allowing the Ca^2+^ entry into the neuron [64]. LTP and LTD both induce the activation of different transduction pathways, such as Ca^2+^/calmodulin-dependent protein kinase (CaMKII) and adenylate cyclase [82,83]. Through this signaling cascade in LTP, the Protein Kinase A will phosphorylate the AMPAR resulting in an increase in the membrane depolarization of the neuron. Furthermore, NMDAR-dependent LTP implicates lateral diffusion and changes in AMPAR proportions in extrasynaptic and synaptic compartments in the hippocampus [90]. Regarding the NMDAR-dependent LTD, various NMDAR subtypes can trigger LTD depending on various factors such as the induction protocol that is employed, expression levels (which vary according to brain region and developmental stage), and environmental conditions (for example, access to a running wheel) [91]. A pioneering study has shown that glutamate binding to NMDAR could induce a conformational change in the cytoplasmic domain of NMDAR that triggers downstream signaling, resulting in LTD independent of Ca^2+^ influx and so revealing the role of metabotropic NMDARs. Indeed, a low-frequency stimulation produced LTD in the presence of either MK-801 or 7-chlorokynurenate (7-CK), two antagonists abolishing NMDAR ion flux without affecting the glutamate binding site, but not APV, which is a competitive GluN2 antagonist blocking the glutamate binding site [53]. NMDAR functions are also the products of the LTP and the LTD processes. The LTP affects the distribution of NMDARs at the synapse, with, for example, in the hippocampal CA1 of adult animals promoting the rapid insertion of GluN2A-containing NMDARs [64]. On the contrary, LTD involves the release of Ca^2+^ from IP3-sensitive intracellular stores and is expressed via the internalization of NMDARs [88]. 

#### 1.1.4. NMDARs in Neuronal Death and Survival

Synaptic NMDARs are suggested to be preferentially neuronal survival promoters in physiological conditions, whereas extrasynaptic NMDARs are notoriously involved in neuronal death whenever excessive and persistent glutamatergic stimulation is present. Synaptic NMDAR pro-survival roles implicate three well-known pathways: i/oxidative stress protection; ii/suppression of the pro-apoptotic p53 upregulated modulator of apoptosis [60]; iii/activation of the transcription factor cAMP response element-binding protein (CREB) and production of the pro-survival protein brain-derived neurotrophic factor (BDNF) [92]. Ca^2+^ influx through NMDARs is not only fundamental for normal physiological processes in neurons but also plays a major role in excitotoxicity. Excitotoxicity is defined by an excess of glutamate release, both on extra- and synaptic compartments, resulting in an over-activation of NMDARs, leading to Ca^2+^ overload inside neurons [93,94]. The Ca^2+^ overload, both on extra- and synaptic compartments, triggers a range of downstream pro-death signaling events such as calpain activation, reactive oxygen species (ROS) generation, and mitochondrial damage, inducing cell necrosis or apoptosis [95]. The excitotoxicity is also described as the first phase of neuronal insults in stroke and traumatic brain injury (TBI) [96,97] before being linked with other CNS diseases [95,96,98,99]. NMDAR distribution on the neuronal surface and their subsequent roles are a complex process. First of all, NMDARs explore both synaptic and extrasynaptic compartments in a minute time frame [61]. In the postsynaptic density (PSD) area, surface NMDARs are dynamically anchored through the interaction between GluN2 subunits and PDZ (Postsynaptic density 95; Discs large, Dlg; Zonula occludens-1)-binding domain proteins and can move to the extrasynaptic compartment. Moreover, synaptic and extrasynaptic NMDAR functions are required for the induction of pro-survival and pro-cell death signaling. The level of excitotoxicity is modulated by the magnitude and duration of both synaptic and extrasynaptic NMDAR activation [100]. Interestingly, tPA is also described as an NMDAR modulator. By binding to vascular and neuronal NMDAR subunit GluN1, tPA modulates NMDAR signaling, leading to the control of the survival of neurons [45,101,102,103,104].

### 1.2. NMDARs on Other Cell Types: Structure, Distributions, and Functions

#### 1.2.1. NMDARs on Endothelial Cells and Tight Junctions of the BBB

The endothelium is a layer of cells linked by tight junctions such as occludin, claudin 5, or VE-cadherin, lining the inner walls of all vasculatures and controlling the passage of leukocytes or molecules between tissue and blood. In 1998, the first observation of NMDAR-subunit GluN1 in cultured neuro-endothelial cells was discussed [45] (Table 2). In the past two decades, neurovascular research has consistently confirmed the localization of NMDARs on endothelial cells using transcriptomic techniques. A large body of research is now investigating the association of their presence on the endothelium to their functionality in regulating the blood-brain barrier (BBB) [105,106,107,108]. 

In the vascular compartment, NMDARs are expressed at the luminal surface of the endothelial cells and close to the tight junctions [109,110]. Endothelial NMDARs (eNMDARs) are likely important regulators of blood-brain and blood-spinal cord barriers maintenance and permeability, oxidative stress, neurovascular inflammatory processes, immune cell transmigration, mitochondrial function, nitric oxide (NO) generation and are part of the neurovascular unit (NVU) [108,111,112,113,114]. The NVU is considered the smallest signaling module in the brain that, besides neurons and microvascular endothelial cells, encompasses mural cells as pericytes, astrocytes, and/or smooth muscle cells, depending on the cerebrovascular tree position [115]. The NVU contributes to ensuring adequate energy substrate delivery from blood to brain cells by regulating glucose transport. NMDAR agonists dilate isolated arteries free of neural circuitry by activating endothelial nitric oxide synthase (eNOS). Several studies raise the possibility that NMDARs expressed by the cerebrovascular endothelium could mediate vasodilation directly [106,116], regulate cerebral blood flow (rCBF) and hemodynamic responses in sensory hyperemia *in vivo* [106,113]. Furthermore, the treatment of endothelial cells with NMDA shows a clear decrease in endothelial resistance [108], and the blocking of the interaction of tPA with the GluN1 subunit decreases leukocyte infiltration, preventing inflammatory mechanisms [107,111]. Moreover, it has been reported in stroke that the release of interleukine-6 (IL-6), a marker of inflammation, is enhanced by tPA through activation of NMDARs and upregulation of endothelin-1 (ET-1) and c-Jun N-terminal kinase (JNK). This cytokine impairs cerebrovascular autoregulation and increases inflammation processes [117].

Anfray and collaborators have studied the role of neuro-endothelial NMDARs in the NVU and demonstrated that circulating tPA released by endothelial cells and neurons into the blood during neuronal activity can agonize NMDARs and thus, elicits vasodilation during functional hyperemia [109]. Indeed, tPA influences NO and ROS production in endothelial cells in an NMDAR-dependent mechanism [118]. Moreover, tPA is described to promote inflammatory reactions in the CNS by increasing the permeability of the blood-brain and blood-spinal cord barriers and facilitating monocyte and lymphocyte migration through rat and human models of BBB [107].

NMDAR signaling induces downstream pathway activation, involving Rho proteins such as Ras homolog family member A (RhoA), Ras homolog family member B (RhoB) and Rho-associated protein kinase (ROCK) [111,119,120]. Rho proteins are key molecules interacting with a large number of targets and directly influencing cytoskeleton rearrangement, cell mobility, and cellular contractility that are disturbed during inflammatory conditions. 

#### 1.2.2. NMDARs on Glial Cells: Astrocytes and Oligodendrocytes

Glial cells are the most abundant cells in the human brain and have long been considered passive supporting cells for neurons. An extensive and convincing literature denied this concept showing the active role of glial cells in the development and functions of the CNS. For more than twenty years, an extensive literature has described NMDARs on developing and mature oligodendrocytes and their functions in physiological and pathological conditions such as brain ischemia [121,122]. NMDARs are blocked only weakly by Mg^2+^ and may contain GluN1, GluN2C, and GluN3 subunits (Table 2). GluN1–GluN3A NMDARs seem to predominate in precursor and immature oligodendrocytes and are replaced by GluN1–GluN2 NMDARs in mature oligodendrocytes. NMDARs are present in myelinating processes of oligodendrocytes, where the small intracellular space could promote a large rise in intracellular ion concentration in response to NMDAR activation [71,123]. Similar to neurons, oligodendrocytes are activated by both glutamate and glycine (excitatory glycine only) [124,125] and are also vulnerable to glutamate toxicity. Compared with neurons, oligodendrocytes are mainly composed of the NMDAR subunit GluN3A combined with the GluN1 subunit [126]. Whereas Mg^2+^ brings a strong blockage of neuronal NMDARs, NMDARs of oligodendrocytes are less sensitive to Mg^2+^ blocking [127,128,129]. When the NMDARs present on oligodendrocytes are exposed to agonists, passivation exists during long-term exposure [130]. Furthermore, in the last decade, many studies have shown that NMDARs of oligodendrocytes are implicated in myelin formation and, on the contrary, in excitotoxicity mechanisms [1,131].

In the same period, studies have demonstrated the presence of NMDARs on the astrocyte membrane with low abundance and their ability to evoke intracellular Ca^2+^ increase in a mutually unexclusive ionotropic and metabotropic mechanism [72]. Growing evidence from *in vitro* and *ex vivo* studies confirmed that activation of NMDARs in astrocytes by glutamate or selective NMDAR agonists mediates ion currents and intracellular Ca^2+^ waves. The expression of NMDARs in astrocytes has been reproducibly documented at the level of mRNA coding for the different subunits. Transcripts for all seven NMDAR subunits (i.e., GluN1, GluN2A-D, and GluN3A-B; Table 2) have been found both in cultured human [132] and rat astrocytes [133]. Skowronska and collaborators demonstrated that NMDA treatment induces Ca^2+^ accumulation in astrocytes *in vitro*, and this accumulation is not observed when GluN1 expression is inhibited by a siRNA knockdown approach [72]. However, the Ca^2+^ permeability of NMDARs on astrocytes is lower than on neuronal NMDARs [134]. Conflicting data exist about the role of astrocyte NMDARs on neuronal protection. Indeed, Jimenez-Blasco and collaborators reported that *in vitro* persistent NMDA stimulation on rat astrocytes induces, by a metabotropic-dependent mechanism, the activation of the transcription factor: nuclear factor-erythroid 2-related factor 2 (Nrf2), leading to the expression of antioxidant genes coding glutamate-cysteine ligase, catalytic subunit and heme oxygenase-1 enzymes [135]. In co-culture with NMDA-treated astrocytes, neurons are protected against oxidative damage. In contrast, it was shown that overexposure of mouse astrocytes to NMDA reduces expression of the potassium channel Kir4.1, aquaporin-4, and glutamine synthase, involved in the neuroprotective functions of astrocytes [72]. So, in the context of glutamate accumulation observed in neurological disorders such as stroke, amyotrophic lateral sclerosis (ALS), multiple sclerosis (MS), Alzheimer’s disease (AD), Parkinson’s disease (PD), epilepsy, and TBI, NMDARs on astrocytes could also be responsible for deleterious responses [136,137,138]. 

Pericytes, along with microglia, astrocytes, and neurons, constitute a major brain cell type. Pericytes act as important regulators of brain functions, notably by maintaining the endothelial BBB integrity and avoiding infiltration of neurotoxic proteins, pathogens and leukocytes in the brain [139]. In AD, it has been shown that pericytes play a key role in BBB clearance of Aβ peptides [140]. Furthermore, in post-mortem studies of AD patients, a BBB leakage and a degeneration of BBB-associated pericytes have been demonstrated, supporting their implication in the disease [141]. In a pre-clinical APOE4;5xFAD mouse model of AD, it has been shown that APOE4 induces a loss of capillary pericytes coverage in hippocampus and brain cortex regions, linked by a decrease of ZO-1 and occludin expression in tight junctions between endothelial cells forming the BBB [142]. However, pericytes do not exclusively interact with endothelial cells. Pericytes can respond to neuronal activity via the establishment of neurovascular coupling. The first hypothesis was the induction of a direct signalization between neurons and pericytes through NMDAR-mediated NO production. Stimulated neurons release glutamate, activating metabotropic glutamate receptors localized on astrocytes, which induce a signal transmitted from astrocytes to capillary pericytes [143]. Thus, this is an indirect interaction. Sweeney and collaborators have described in an extensive review of the literature the link between BBB breakdown and neurodegeneration among neurodegenerative disorders including AD, PD, ALS, MS, and other CNS disorders such as stroke, TBI, spinal cord injury, and epilepsy. In this review, the authors highlighted the role of pericytes in AD, HD, and ALS pathophysiologies [144]. However, NMDARs have not been identified on pericytes yet. Without a differentiation step that permits the generation of neural-like cells from pericytes, this cell type is not described to express NMDARs, particularly the NMDAR-obligatory GluN1 subunit [145,146] (Table 2).

#### 1.2.3. NMDARs on Immune Cells: Microglia, Macrophages, and Immunological Synapse

Although less studied, expression of functional NMDARs has also been reported in non-neuronal nor glial cells, in particular in cells from the immune system (Table 2).

T cells are primed through the formation of a stable T cell-antigen-presenting cell (APC) junction, known as the immunological synapse (IS) [6]. IS forms a specialized area with surface receptors that integrates signals from membrane ligands or soluble mediators such as cytokines. It was shown that thymocytes expressed GluN1, GluN2A, and GluN2B subunits at mRNA and protein levels as demonstrated by intracytoplasmic assay and fluorescence-activated cell sorting (FACS) analysis [74]. NMDA treatment did not induce Ca^2+^ signaling in resting or activated thymocytes. Instead, experiments with memantine or MK-801 antagonists showed that NMDA-dependent Ca^2+^ signaling required T cell-dendritic cell (DC) IS. Interestingly, DCs could release glutamate, that act as a signaling molecule and an immune modulator. In another study, Kahlfuß and collaborators confirmed that GluN1 was present at the mRNA level but did not evidence GluN1 protein expression by Western blot or FACS analysis [147]. T cell proliferation and cytotoxicity that were inhibited by memantine or MK-801 were attributed to a blockage of Kv1.3 and KCa3.1 channels but not NMDARs. Thus, further investigations are needed to understand the accurate contribution of NMDARs in T cell biology. 

Gonias and collaborators have described those interactions of the tPA with NMDARs, as being involved in the regulation of macrophage activation by inflammatory stimuli [75,148]. Indeed, binding of tPA with NMDARs inhibited proinflammatory cytokines expression by macrophages stimulated by lipopolysaccharide (LPS). *In vivo*, enzymatically inactive tPA blocked the toxicity of LPS suggesting that targeting tPA-NMDAR interaction could be beneficial to damper pathological inflammation such as in sepsis. 

Microglial cells constitute the resident macrophage subset in the CNS. While the expression of NMDARs on microglia is poorly documented *in situ*, some reports show that microglial cells express a modest level of functional NMDARs *in vitro* [73,149]. Stimulation with NMDA triggers microglial activation and proliferation and contributes to neuron cell death *in vitro* through the production of NO and pro-inflammatory cytokines [73,150]. However, it is still controversial whether NMDARs on microglia could mediate measurable membrane currents [151,152]. 

Some data exists about the function of NMDARs on neutrophils. Activated neutrophils are able to produce both glutamate and D-serine, and pharmacological blockade of NMDAR signaling inhibits ROS production suggesting an autocrine modulation mechanism [76]. So, NMDAR signaling on neutrophils may contribute to enhancing anti-infectious immune response but also worsens tissue injuries through the excessive production of toxic mediators. 

## 2. CNS Diseases and NMDAR Dysfunctions

NMDAR dysfunctions have emerged as a key target in the therapeutic strategies of several major nervous system disorders, including chronic neurodegenerative diseases, traumatic or ischemic brain injury, mental disorders, and diseases with a neurodevelopmental or autoimmune origin. Either hyperactivity or hypofunction of NMDARs could contribute to disease pathophysiology. It is likely that distinct subtypes of NMDARs (as defined by subunit composition, cell type, and location) are differentially involved in CNS diseases.

### 2.1. Neurodegenerative Diseases

#### 2.1.1. Alzheimer’s Disease

AD accounts for approximately 60–80% of all cases of dementia and is neuropathologically characterized by extracellular deposits of insoluble Aβ and intracellular aggregates of hyperphosphorylated tau proteins. Although, for a long time, it was believed that the extracellular accumulation of Aβ was the culprit of the symptoms observed in these patients, more recent studies have shown that cognitive decline in people suffering from this disease is associated with soluble Aβ-induced synaptic dysfunction instead of the formation of insoluble Aβ-containing extracellular plaques [153]. These observations are translationally relevant because soluble Aβ-induced synaptic dysfunction is an early event in AD that precedes neuronal death. Thus, this observation leads to new potential therapeutic interventions to prevent cognitive decline before the progression to irreversible brain damage [154]. As explained above, NMDARs are critical for synaptic plasticity and the survival of neurons. NMDAR functions must be maintained at a physiological level to avoid the toxic events caused by its hyperactivation, such as excitotoxicity, BBB disruption, and ultimately neurodegeneration as occurs in AD. The major factors that affect NMDAR signaling in AD include glutamate availability and the modulation of ionotropic and metabotropic NMDAR functions [155]. In AD patients, a decrease in glutamate transporter capacity along with their protein expression and a selective loss of vesicular glutamate transporter (VGluT) have been described [156]. In parallel, the astrocyte excitatory amino acid transporter 2 (EAAT2), which is located close to the synaptic cleft, was reported to have impaired function in AD [157]. Moreover, the accumulation of Aβ peptide disturbed the glutamate release machinery causing the long-term reduction of synaptic glutamatergic transmission and the inhibition of synaptic plasticity [44]. Pathologically, elevated levels of Aβ peptide secreted in the extracellular space may also indirectly induce a partial blockage of NMDARs and shift the activation of NMDAR-dependent signaling cascades toward pathways involved in the induction of LTD and synaptic loss [44]. Liu and collaborators summarized the literature indicating that NMDARs are i/an important downstream target of Aβ; ii/necessary for the modulation of Aβ-dependent synaptic perturbation and loss; iii/potentially a receptor for Aβ and iv/involved in the formation of Aβ [44]. Finally, the astrocytic NMDARs may contribute to AD due to their roles in facilitating glutamate excitotoxicity [154]. Memantine, a moderate NMDAR channel blocker which has been approved by the European Medicines Agency (EMA) in 2002 and the food Drug Administration (FDA) in 2003 for the treatment of moderate to severe AD, highlighted the importance of NMDARs in AD physiopathology [158]. Memantine has been described to preferentially inhibit extrasynaptic NMDARs [159,160], reinforcing the pathological role of an overactivation of these receptors [44,154].

#### 2.1.2. Parkinson’s Disease

PD is a progressive degenerative nervous system disorder and is the second most common neurodegenerative disorder in the elderly population. The causes of PD are unknown. However, there is evidence supporting the involvement of the immune response and excitotoxicity in the degeneration of dopaminergic (DA) neurons [137,161]. The disease originates from the loss of dopamine-producing neurons in the substantia nigra (SN) in the brain, resulting in the unregulated activity of the basal ganglia. A BBB disruption has been noted in animal models of PD and forms a basis of the vascular hypothesis of neurodegeneration [162]. Alpha-synuclein (α-syn) is a protein found to aggregate in the substantia nigra region of patients with PD, forming Lewy Body inclusions. Its aggregation may contribute to neuronal cell death in PD [163]. Dopaminergic drugs, including the dopamine precursor levodopa (l-3,4-dihydroxyphenylalanine, L-DOPA) and dopamine receptor agonists, are currently considered the only standard therapy for treating Parkinsonian symptoms. In PD, the reduction of striatal dopamine (DA) release due to the degeneration of dopaminergic afferents generates multiple alterations of the synaptic physiology, and mainly affects the striatal glutamatergic transmission, especially in spinal projection neurons (SPNs) [164]. Animal models of PD demonstrated that L-DOPA treatment induced an increase in glutamate transmission and is implicated in the development and maintenance of these motor complications [165], hyperphosphorylation of striatal NMDAR subunits, and upregulation of both NMDARs and AMPAR [166]. The composition and the location of NMDARs are important in the neurotoxic effects observed in PD as the GluN2B subunit has been described to be heavily distributed in the striatum and other basal ganglia regions [167]. Furthermore, a pathological redistribution of GluN2A and GluN2B subunits at the synapse is correlated with motor abnormalities in L-DOPA-treated dyskinetic rats [168]. A chronic L-DOPA treatment increased the expression of the GluN2D subunit in the striatal projection neurons [169], a subunit linked with potentiation of NMDA-dependent excitotoxicity by tPA [102,159]. Interestingly, intrastriatal α- Synuclein Preformed Fibrils (α-syn- PFF) also lead to glutamatergic system overactivity and striatal alterations as a high dose can impair SPN neuronal activity by reducing GluN2A-dependent NMDAR functions [170]. Both PD-induced dopamine depletion and L-DOPA treatment led to the redistribution of NMDAR subunits in L-DOPA-treated dyskinetic rats and monkeys’ models. Clinical treatment with NMDAR modulators for PD patients is limited even if a growing literature shows the rationale of developing this strategy with more specific modulation of NMDAR signaling [12,171]. Interestingly, tPA has been reported in the substantia nigra where it could lead to excitotoxic and pro-inflammation events [117,167,172,173].

#### 2.1.3. Huntington’s Disease

HD occurs worldwide with a prevalence of ~12 per 100,000 individuals in populations of European descent. This disease is caused by a dominantly inherited CAG repeat expansion in exon 1 of the huntingtin gene (HTT). It is characterized by progressive involuntary choreiform movements, behavioral and psychiatric disturbances, and dementia. The onset of the motor symptoms of HD, known as motor onset, can occur from childhood to old age, with an average age of 45 years, and is followed by inexorable disease progression [174]. Purkinje neurons present in the cerebellum express NMDARs [175] and are a sensitive and specialized cell type important for fine motor movements and coordination. Purkinje cell damage manifests as motor incoordination and ataxia, a prominent feature of many human disorders, including spinocerebellar ataxia and HD.

NMDARs are highly expressed in striatal medium spiny neurons (MSNs), the major neuronal population in the striatum that degenerates in HD [7]. Increased levels of extrasynaptic NMDARs in MSNs are seen in an HD mouse model (YAC128), and their activation appears to contribute to the vulnerability of MSNs to excitotoxicity caused by mutant Huntingtin protein (mtHTT) [176]. Extrasynaptic NMDARs contain GluN2B subunit, which contributes more to the total NMDA-evoked current in D2 dopamine receptor-containing MSNs than in D1-containing MSNs in an HD mouse model. These observations are consistent with the earlier degeneration of D2-MSNs in HD [177]. Furthermore, crossing GluN2B-overexpressing mice with mice from an HD model exacerbates the death of MSNs. Thus, extrasynaptic GluN2B-NMDARs could play an important role in neuronal cell death in HD [178]. 

#### 2.1.4. Amyotrophic Lateral Sclerosis

ALS leads to the neurodegeneration of upper and lower motor neurons, progressive muscle weakness, and death due to respiratory failure [179,180]. The cumulative lifetime risk of ALS is ~1 in 300 [181], and the incidence is 1–2 per 100,000 person-years [182]. Median survival after symptom onset in patients is 3–5 years. About 10% of ALS cases are familial, with the remaining 90% sporadic. There is no effective treatment for ALS except the benzothiazole kilomole (Riluzole), which modestly slows disease progression allowing an extension of survival [183]. Identification of mutations in the familial form helps to elucidate the mechanisms involved in ALS. Among these mutations, a coding one has been identified in the D-amino acid oxidase (DAOR199W) [184] and is associated with an impairment of D-serine metabolism, causing protein aggregation, autophagy, and cell death in motor neuron cell lines [185]. Using an *in vitro* DAOR199W expressing cells model and a selective antagonist at the glycine/D-serine binding site of the NMDARs (5,7-dichloro-4-hydroxyquinoline-2-carboxylic acid; DCKA), Paul and de Belleroche highlighted an NMDAR-dependent increase in LC3-II and autophagic-mediated cell death and reported dysregulation of glycine and D-serine, both endogenous co-agonist of NMDAR, observed in several preclinical ALS studies [186,187] (Table 1). Recently, an elevated amount of D-serine has been observed in the plasma of ALS patients [188]. Furthermore, modifications of Zn^2+^ amounts, another endogenous modulator of NMDARs [20] (Table 1), are observed in a mouse model [superoxide dismutase 1 transgenic (SOD1 Tg) mouse] of ALS at different stages of disease development [189]. Interestingly, on induced motor neurons (iMNs) from C9ORF72 ALS/FTD patients, elevated cell surface levels of NMDARs and AMPARs are found on neurites and dendritic spines compared with control iMNs and might induce hyperexcitability and cell death due to this increased glutamate activation [190]. As a reminder, Riluzole inhibits Kaïnate and NMDAR-dependent currents [191] by controlling neuronal hyperexcitability, decreasing the voltage-dependent sodium channels, and reducing glutamate release, uptake, and glutamate receptor function [192]. As in other neurodegenerative diseases, the physiopathology of ALS involves inflammation mechanisms and a number of associated proteins. As in in many other neurodegenerative diseases, the pathophysiology of ALS involves inflammatory mechanisms and numerous associated proteins. Recently, in an extensive review of the literature, O’Day and Huber have reported that calmodulin is implicated in neuroinflammation through its interactions with protein complexes, including NMDARs [193]. D-serine is a well-known co-agonist of NMDARs on the glycine binding domain, providing insight into potential upstream mechanisms that involve NMDARs in neurodegeneration [184,185].

### 2.2. Neurovascular and Traumatic Disorders

#### 2.2.1. Stroke

Stroke is a leading cause of death and disability in humans. Every 40 s, someone in the United States has a stroke. Every 3.5 min, someone dies of a stroke [194]. According to American Heart Association, the global prevalence of stroke in 2019 was 101.5 million people, whereas that of ischemic stroke was 77.2 million, that of intracerebral hemorrhage was 20.7 million, and that of subarachnoid hemorrhage was 8.4 million. Stroke is also the principal cause of long-term disability irrespective of age, sex, ethnicity, or country and was responsible for 116 million years of life lived with disability in 2016. Intracerebral hemorrhages are the deadliest form of acute stroke and are defined by brain injury attributable to acute blood extravasation into the brain parenchyma from a ruptured cerebral blood vessel [194]. An ischemic stroke happens when the brain’s blood vessels become narrowed or blocked by fatty deposits that build up in blood vessels or by blood clots or other debris that travel through the bloodstream, causing severely reduced blood flow.

Over-activation of NMDARs after stroke induces an excessive increase of intracellular Ca^2+^, a process leading to neuronal death [1,20,195]. In stroke, NMDAR-dependent excitotoxicity appears to be a primary cause of neuronal death occurring acutely after ischemia or injury (short window, around 1 h) [196], and NMDAR blockers or modulators protect neurons against ischemic cell death *in vitro* and *in vivo* [101,197,198,199]. Interestingly, NMDAR excitotoxicity also seems to be subunit-dependent since selective GluN2B or GluN2D antagonists blocked, while GluN2A-preferring antagonists exacerbated, ischemic cell death [102,199]. Thus, it is possible that excessive activation of GluN2B-NMDARs underlies ischemic cell death, whereas the activity of GluN2A-NMDARs may promote recovery after the ischemic insult. By an overactivation of the NMDAR signaling, the increase of the endogenous level of tPA or the injection of recombinant tPA (rtPA) promotes Ca^2+^ influx and the subsequent excitotoxic neuronal death [45,102,103,159,200]. The BBB integrity is disrupted after stroke onset, and tight junctions between endothelial cells disappear, allowing leukocyte infiltration into the injured brain tissue [201]. When expressed on brain endothelial cells, NMDARs are involved in the maintenance of the integrity of the BBB [118]. Zhang and Yepes’s group revealed the role of tPA in microglial activation in an ischemic brain injury model [202]. The blocking of the interaction of tPA with the GluN1 subunit decreases leukocyte infiltration and prevents inflammatory mechanisms [107,109]. Concerning hemorrhagic stroke, except for “last chance” invasive approaches, no drug treatment is available [203]. Despite the great incidence, the severity of the outcomes, and the high economic costs of ischemic stroke, rtPA is the only treatment approved by the FDA or EMA, and the mechanical thrombectomy has demonstrated beneficial effects on a selected population of patients. However, due to safety concerns such as the risk of hemorrhagic transformations after treatment with rtPA combined with its NMDAR-dependent-excitotoxic and pro-inflammatory effects, the number of patients who can use this drug is very low [45,204]. Interestingly, the balance between nocuous and beneficial effects of tPA also depends on its local concentration: a low concentration of tPA (up to 10 nM) protects neurons against excitotoxin-induced cell death, whereas a high concentration (300 nM) induces cell death [200,205,206]. In addition, even when blood flow is restored, secondary damages caused by reperfusion can be observed in brain tissue, mainly because of the production of deleterious substances such as ROS and inflammatory cytokines [207]. The disruption of the BBB and inflammation within the first hours after ischemia may represent a crucial step of reperfusion injury and consequent hemorrhagic transformation [208]. 

#### 2.2.2. Traumatic Brain Injury

TBI has been one of the leading causes of morbidity, disability, and mortality across all ages, with more than 50 million individuals suffering from TBIs each year [209,210,211]. Globally, more than 3 million TBI survivors experience post-traumatic complications ranging from neurological and psychosocial problems to long-term disability [212,213]. Damages of neuronal tissues associated with TBI fall into two categories: (i) primary injury, which is directly caused by mechanical forces during the initial insult; and (ii) secondary injury, which refers to further tissue and cellular damages following primary insult [214]. As in stroke, NMDAR-dependent excitotoxicity is a primary cause of neuronal death occurring in the injury [196]. The primary injuries can be focal or diffuse and be simultaneously present in patients who suffer from moderate to severe TBI. The necrotic area of neuronal and glial cells is concentrated at the impact side with compromised blood supply, causing the occurrence of hematoma, epidural, subdural, and intracerebral hemorrhages at confined layers of the brain. The strong tensile forces damage neuronal axons, oligodendrocytes, and blood vasculature, leading to brain edema and ischemic brain damage [214]. The endogenous and exogenous tPA protects against white matter injury after TBI without increasing intracerebral hemorrhage volumes but have toxic effects on the gray matter [215,216]. Endogenous tPA is overexpressed around the hematoma [217,218]. Within the acute post-TBI period of 24 h, dysfunction of BBB allows infiltration of circulating neutrophils, monocytes, and lymphocytes into the injured brain parenchyma, processes involving NMDARs [111,219]. Analysis of cerebrospinal fluid and post-mortem tissue of TBI patients revealed that these polymononuclear leukocytes release complement factors and pro-inflammatory cytokines [214]. Mechanistically, several factors contribute to secondary injuries, which include neuroinflammation, excitotoxicity, oxidative stress, mitochondrial dysfunction, lipid peroxidation, axon degeneration, and apoptotic cell death [220]. The initial focus on the role of NMDARs in TBI was based on the observation that high concentrations of extracellular glutamate were present after trauma, which results in an excessive influx of Ca^2+^ via NMDARs and subsequent activation of intracellular signaling pathways, including CAMKII [221], PKA [222], PKC [223], MAPK [224] or protein phosphatases involved in neural injury and death [97,121]. An interesting study by Singh and collaborators aimed to determine the mechanosensitive nature of GluN2B-containing NMDARs and the signaling cascades that are involved in the regulation of NMDARs’ response to mechanical stimuli [225]. However, another study claimed that the GluN2A subunit-containing NMDARs displayed a greater response [226].

### 2.3. Autoimmune Diseases

#### 2.3.1. Multiple Sclerosis

MS is a chronic disease of the CNS that usually first manifests between ages 20 and 40 and is estimated to affect about 2.5 million people worldwide. MS is a demyelinating disease of the CNS, characterized by an immune infiltration into the CNS, inflammation, demyelination, and finally, axonal degeneration. The cause of MS is not completely understood, but recently a longitudinal study with a cohort of 10 million adults demonstrate that an Epstein-Barr virus (EBV) infection is a key event for the development of MS [227]. In MS, the immune system attacks the protective myelin sheath that covers nerve fibers and causes motor deficits. Eventually, the disease can cause the nerves themselves to deteriorate or become permanently damaged, leading to neurodegeneration [228]. Four disease courses have been identified in multiple sclerosis: clinically isolated syndrome (CIS), relapsing-remitting MS (RRMS), primary progressive MS (PPMS), and secondary progressive MS (SPMS).

Histopathological studies have implicated dysregulation of the glutamatergic system in the pathogenesis of MS, and animal studies have helped to decipher the mechanisms by which excessive glutamate might contribute to the disease process [114]. In an animal model of experimental autoimmune encephalomyelitis, NMDARs are involved in the BBB permeability and control of leukocyte infiltration, preventing the progression of neurological impairments [107]. NMDARs are expressed in the vicinity of neuro-endothelial tight junctions of the blood–spinal cord and BBB. The tight junction-associated spinal cord endothelial NMDARs contain GluN1, GluN2B, and GluN3A subunits [111]. The intracellular pathways that mediate the toxic effects of NMDAR activation in endothelial cells include the production of ROS and NO, increased intracellular Ca^2+^ [108], and activation of intracellular kinases such as RhoA kinase [111,119]. On the other hand, excessive glutamate release can lead to synaptic damage and excitotoxicity in neurons coupled with a demyelination process, possibly by inducing excitotoxic death in myelinating oligodendrocytes [114].

#### 2.3.2. Anti-NMDAR Encephalitis

Anti-NMDAR encephalitis is a rare neurological disorder with an autoimmune etiology. Worldwide, 1–10 per 1 million people are diagnosed each year [229], with a strong sex ratio in favor of men (women: men ratio, 4:1) and a median age of diagnosis of 21 years old [230]. Anti-NMDAR encephalitis was first described as associated with ovarian teratomas in women [231], but this initial observation has been reassessed to be less than first thought, with 58% of diagnosed women concerned [232]. With tumors (mostly ovarian carcinoma), infection by herpes simplex virus (HSV) [233] is the second leading trigger of anti-NMDAR encephalitis onset. Despite these two different classes of etiologies, both lead to a common symptom pattern and pathophysiology. A prodromal phase with psychiatric symptoms following neurological signs and sometimes death if not treated are described [230,234]. The diagnosis is based on the major clinical symptoms such as behavioral modifications, movement abnormalities, and seizures. The presence of antibodies against the GluN1 subunit of NMDARs in a patient’s cerebrospinal fluid (CSF) is the major biological parameter [235,236].

Mechanistically, the outbreak of the disease is caused by the presence of anti-NMDAR autoantibodies produced by dysregulated B lymphocytes [237,238]. One consequence of this autoantibody’s production is the hyporesponsiveness of NMDARs on neurons due to their internalization after binding [234,239]. It has been experimentally demonstrated that immunization of mice with NMDAR proteins leads to the production of anti-NMDAR antibodies and the symptomatology of anti-NMDAR encephalitis in mice [240]. This provides a strong proof of concept that the pathophysiology of NMDAR encephalitis is, at least in part, caused by the involvement of autoantibodies. Additionally, in a more recent mouse model of this disease, a B cell epitope of the GluN1 subunit of NMDARs was identified as an immunogenic amino acid sequence able to induce encephalitis-like symptoms [238]. These findings highlight a new potential therapeutic target, using NMDAR antagonism strategies [241,242].

### 2.4. Mental Diseases

#### 2.4.1. Depression

Major depressive disorder (MDD) is a common mental disorder and is one of the leading causes of disability worldwide. The clinical features are depressing-mood, pessimism, and world-weariness [243]. Although the exact cause is unknown, several lines of evidence suggest that MDD in most people is caused by a combination of genes and stress, which can change brain chemistry and reduce the ability to maintain mood stability. Genetic factors, including genetic variants within genes that operate in stress response mediating neurobiological systems, neurotransmitters, and synaptic plasticity, can increase susceptibility for MDD [244]. Dysfunction of glutamatergic neurotransmission in the brain’s cerebral cortex and limbic region is associated with MDD. The alterations in NMDAR subunits, especially the GluN2A, and GluN2B, as well as PSD-95, suggest an abnormality in NMDAR signaling in the prefrontal cortex in patients with major depression [245]. The amount of GluN2A, GluN2B, and PSD-95 immunoreactivity from depressed subjects was also significantly lower (−54%, −48%, and 40%, respectively) than that of control subjects. The results might reflect a hypofunction of NMDARs due to the important role of PSD-95 in the trafficking, membrane targeting, and internalization of NMDAR complexes. Thus, lower levels of PSD-95 may reflect reduced communication/coupling of NMDARs with intracellular signaling cascades. Moreover, Wang and collaborators reported in their review that the hypermethylation of the GRIN2A gene, which encodes for the GluN2A subunit, was detected in the prefrontal cortex (PFC) and hippocampus of MDD patients leading to the abnormal expression of GluN2A and increasing the susceptibility to depression. They also highlighted that a knockout of the mouse *GRIN2A* gene reduces their anxiety and depression-like behaviors. When the *GRIN2B* gene that encodes the GluN2B subunit was selectively knocked out of the principal cortical neurons, synaptic protein synthesis and mTOR activation were significantly increased, together with relieving depression-like behaviors. *GRIN2B* gene polymorphism can predict treatment resistance to MDD and suicide attempts. A genome-wide association study in European samples has demonstrated that rs220549 in GRIN2B is associated with depression, suggesting *GRIN2B* may be a promising candidate gene for MDD [243]. Finally, several animal studies have shown that antidepressants could alter the brain region-specific expression of NMDARs [246]. NMDAR antagonist ketamine produces rapid and long-lasting antidepressant actions in treatment-resistant patients [247]. A systematic review and meta-analysis study found that ketamine rapidly reduced suicidal thoughts within one day and for up to one week in depressed patients with suicidal ideation, suggesting a potential rapid-acting treatment for patients at risk of suicide [248]. Extensive investigations on ketamine’s role in depression will keep the door open for developing more effective NMDAR antagonists as antidepressants [245,246]. 

#### 2.4.2. Schizophrenia

SCZ is a severe psychiatric disorder affecting nearly 22 million people around the world [249] with positive symptoms (delusions, hallucinations, and disorganized speech and behavior), negative symptoms, and cognitive impairment. Their origins seem to lie in genetic and/or environmental disruption of brain development, including in a hypofunction of NMDARs [86,250]. Indeed, approximately 70% of diagnosed patients present a genetic heritability, with a recent highlighting concerning 287 distinct genomic loci concentrated in genes expressed in excitatory and inhibitory neurons of the CNS, implicated in synaptic organization, differentiation, and transmission. One of these genes is the one encoding for the GluN2A subunit of NMDARs [251]. This finding supports the neurobiological hypothesis of the implication of NMDARs in the pathophysiology of SCZ and could explain the NMDAR hypofunction. Moreover, to support the hypofunction of NMDARs, the administration of NMDAR antagonists constitutes a model of SCZ [236]. Several studies demonstrated that in the serum of patients suffering from SCZ, anti-NMDAR autoantibodies are found [252,253]. Moreover, it has been demonstrated that the seroprevalence of anti-NMDAR G-type immunoglobulins (IgG) was significantly higher in patients compared with controls [254]. Nevertheless, the presence of anti-GluN1 IgG, which are the only ones whose pathogenicity has been demonstrated [255], remains rare for all schizophrenic patients. The only study based on gold standard methods did not find any patient positive for anti-GluN1 IgG [256]. Studies have shown that SCZ patients have lower tPA and higher PAI-1 levels than the general population [257]. In the psychiatric field, the link between psychotic disorders, particularly SCZ and immune system dysregulations, including autoimmunity, is a concept that regained strong support thanks to the better characterization of inflammatory-induced psychotic symptoms and autoimmune encephalitis, the most well characterized of which is anti-NMDAR encephalitis [258].

### 2.5. Neurodevelopmental Diseases

#### 2.5.1. Autism

Autism spectrum disorder (ASD) is a complex neurodevelopmental disorder characterized by the core symptoms of social deficits, language communication failure, and stereotyped behaviors. The current prevalence of ASD is estimated to be 1.5% or higher in developed countries. However, the etiology of autism remains largely unexplained [259]. ASD exhibits a strong genetic basis, and increasing evidence from inherited and de novo gene variations suggest a notable convergence on synapse pathophysiology in ASD, particularly dysfunction of excitatory synaptic transmission. Autism and SCZ have some overlapping features, such as similar disturbed cognitive and social functions, neurobiological (brain volumes), and genetic (e.g., involvement of the same genes or chromosomal locations) domains [260]. Therefore, researchers apply the “neuroinflammation hypothesis of ASD” to regard autism as it was the same in 1980 for SCZ and the “immune hypothesis of schizophrenia” to explain immune-dysfunction-induced neuroinflammation [261]. At least 69% of individuals with a diagnosis of ASD have been known to have neuroinflammation or encephalitis. Specifically, the so-called “anti-brain autoantibody,” including anti-NMDARs, may damage fetal or children’s brain cells, eventually leading to children falling into an autistic or regressive state [261]. Furthermore, a fragile balance between excitation and inhibition (E/I balance) in synaptic inputs to a neuron and neural circuits is important for normal brain development and function. Accordingly, the disturbed E/I balance [262,263] is linked with NMDAR dysfunction and is observed in ASD. Loss-of-function mutations in the *GRIN1*, *GRIN2A,* and *GRIN2B* genes were found in patients suffering from a variety of neurodevelopmental disorders ranging from childhood encephalopathy to intellectual disability or autism [264,265]. tPA is expressed at a high rate in various brain structures, including the amygdala, hippocampus, and cerebellum [266,267]. An interesting study published in 2016 showed an elevated level of tPA and some adhesion molecules in the serum of children with ASD [268]. 

#### 2.5.2. Epilepsy

Epilepsy has emerged as a global health concern affecting around 65 million worldwide. Despite the rapid progression in clinical and pre-clinical epilepsy research, the pathogenesis of epilepsy remains elusive. Epilepsy affects all age groups and is one of the most common, most disabling neurological disorders, characterized by recurrent seizures [269]. Seizures are characterized by brief episodes of involuntary shaking that may involve a part or the entire body and sometimes are accompanied by loss of consciousness and loss of bowel or bladder control. While the classification of epilepsy is often evolving clinically, epilepsies tend to have two broad categories: focal and non-focal epilepsy. About 25% of all patients with epilepsy have drug-resistant epilepsy (DRE), also called medically refractory or pharmaco-resistant epilepsy [270]. Seizures in this population are reported in up to 50% of cases. 

Evidence demonstrates that BBB breakdown may induce epileptic seizures, and conversely, seizure-induced BBB disruption may cause further epileptic episodes [271]. As an example, Aquaporin-4 channels reduced the buffer of extracellular potassium and facilitated NMDAR-mediated neuronal hyperexcitability and epileptiform activity [272]. Ketamine is a non-competitive NMDAR antagonist. Oral ketamine or an intravenous infusion of a low dose of ketamine as an alternative treatment for RSE in adults has already been used with patients [273]. Epileptic seizures have been described in anti-NMDAR autoimmune encephalitis [229]. 

## 3. Modulators of NMDARs for the Treatment of Neurological Disorders: Preclinical and Clinical Data

For the last three decades, NMDARs have been a privileged target for the treatment of CNS disorders. The discovery of endogenous molecules involved in NMDAR pathways, as well as the development of new pharmacological compounds, have been helping to understand the fragile balance between pro-survival and pro-death NMDAR functions (Figure 1). The classification of these modulators is continuously refined by new data from clinical trials or the repositioning of approved molecules. In parallel, alternative therapeutic approaches targeting NMDARs give new hope for the treatment of pathologies of the CNS and the improvement of patients’ life (Figure 1). 

### 3.1. Channel NMDAR Blockers and Modulators

**MK-801**. Dizocilpine, also called (5 R, 10S)-(+)-5-Methyl-10,11-dihydro-5 H-dibenzo[a,d]cyclohepten-5, 10-imine hydrogen maleate or MK-801 is a potent, selective and non-competitive antagonist of NMDARs discovered in 1982. MK-801 binds inside the ion channel of NMDARs, thus preventing the ions’ flow, including Ca^2+^, through the channel. It blocks NMDAR function in a use- and voltage-dependent manner since the channel must be open for the molecule to be able to bind inside [9]. Since its discovery, MK-801 has been extensively studied for the use in the treatment of diseases with excitotoxic components such as stroke [274] or TBI [214] and neurodegenerative diseases such as HD, AD [44], and ALS [275]. MK-801 has shown effectiveness in protecting neurons in cell culture and animal models of excitotoxic neurodegeneration [276,277,278,279]. The development of MK-801 as a therapeutic drug has been dropped, mainly due to the side effects coming from its stringent ON/OFF mechanism for the blockade of NMDARs in preclinical models [280,281,282]. However, it is still widely used as a research tool or a comparative molecule for the design of new drugs.

**Aptiganel** (cerestat; CNS-1102), another non-competitive potent channel blocker, has shown positive neuroprotective effects in stroke animal models [283,284]. However, aptiganel showed very limited efficacy in a phase II ischemic stroke clinical trial, associated with potential mortality and strong side effects such as elevation of blood pressure and hallucinations [10,285]. This anti-NMDAR drug has failed in nested phase II/III randomized controlled trials in acute ischemic stroke [10]. It is suspected that this result may be linked to NMDAR antagonist interferences with regeneration and repair mechanisms [286].

**Dextromethorphan** (AVP-923) is metabolized *in vivo* to dextrorphan and is most commonly known as a cough suppressant. It is one of the oldest non-competitive FDA-approved NMDAR ion channel blockers. In a 2 h transient middle cerebral artery occlusion (tMCAO) rabbit model, dextrorphan showed significantly reduced cortical and striatal ischemic neuronal damage but did not improve ischemic cortical edema. With four hours delay of occlusion, dextrorphan not only revealed no significant neuroprotection but also worsened the area of ischemic edema [11]. On the other hand, dextromethorphan was investigated in clinical trials for the treatment of pain and depressant effects in several pathological features such as fibromyalgia (NCT03538054 and NCT05068791), major depressive disorder (NCT05181527), AD, HD-related irritability (NCT00788047) as well as an adjunct analgesic for postoperative pain. By acting directly on post-synaptic NMDARs, dextromethorphan attempts to modulate glutamate neurotransmission, thereby reducing targeted symptoms. Furthermore, in an acute ischemic stroke phase II trial, lack of neuroprotective effects, increased myocardial infarctions, and renal failure were allocated to this molecule [287].

**Amantadine**. A low-affinity non-competitive NMDAR antagonist with rapid blocking channel kinetics called amantadine could ameliorate several clinical symptoms in PD, and the chronic treatment might improve apathy and fatigue in patients [12] (NCT00632762). For Huntington’s chorea, amantadine treatment delivered no beneficial effects, but patients felt improvement [288]. However, this molecule shows a beneficial effect on patients with L-DOPA-induced dyskinesia [289]. Concerning the recovery of ischemic and hemorrhagic strokes, a phase II clinical trial is currently underway (NCT05140148). The efficacy of amandine has been also evaluated in patients with TBI in several clinical trials (NCT04527289, NCT00970944).

**Memantine** is derivate from amantadine [290]. Memantine has been shown to exert preferential activity toward extrasynaptic NMDARs. The over-activation of extrasynaptic NMDARs linked to neurodegeneration in AD has also been supported by the first pharmacotherapeutic use of memantine [159,160]. The use of memantine in AD, PD, SCZ, bipolar disorder, and MDD clinical trials revealed no effect or inconsistent efficacy [247,291,292,293,294,295]. A published pilot, open-label, randomized clinical trial considered memantine as a neuroprotective agent in patients with mild to moderate ischemic stroke, based on its significant effects on reducing brain damage (a significative decrease of circulant MMP-9, a neuronal damage biomarker) and improving the neurologic function of the patients [296]. Nevertheless, memantine is currently investigated in subpopulations of patients suffering from neurovascular, neurodegenerative, or neurological disorders such as middle-to-old aged bipolar II disorder patients (NCT04035798), ischemic stroke (NCT02535611), PD (NCT03858270), vascular dementia (NCT0398642), High-Functioning ASD (NCT03553875), SCZ (NCT03860597), motor neuron disease (MND-SMART; NCT04302870) or patients suffering from cognitive impairments. Further, there is growing evidence for its potential efficacy in diminishing side effects in brain tumor therapies (NCT04804644; NCT04567251; NCT04588246; NCT02635009). 

**Ketamine**. There are two enantiomer forms of ketamine, (R)- and (S)-ketamine, with differential pharmacokinetic properties. (S)-ketamine was investigated in several patient subpopulations and has been recorded in clinical trials for the treatment of subpopulation depression syndromes and fibromyalgia. Rapid onset of robust antidepressant effects was observed in patients with treatment-resistant depression after an intravenous infusion of either 0.2 or 0.4 mg/kg of (S)-ketamine. The lower dose may allow for better tolerance while maintaining efficacy [225]. Nevertheless, the administration of high doses of ketamine or phencyclidine (PCP) to healthy volunteers could recapitulate the positive, negative, and cognitive symptoms of SCZ [297,298]. Globally, in the last five years alone, more than 100 phase III clinical trials were initiated using ketamine and its derivates. It is very often used as a combined therapy to investigate therapeutic benefits in postoperative analgesia, suicide threat, postpartum depression, treatment-resistant depressive disorder, cognitive dysfunction, SCZ, and alcohol abuse using a multitude of dosing regimens. Furthermore, ketamine has been approved and is currently used in general anesthesia and short-term sedation [299]. 

**Methadone** is available for clinical use to treat moderate to severe pain and opioid dependence due to its μ-opioid receptor agonism [14]. Dextromethadone (D-methadone/REL-1017), a non-competitive NMDAR antagonist, provided rapid and sustained antidepressant actions via mTORC1-mediated synaptic plasticity in the prefrontal cortex in animal models [300]. As dextromethadone performs as a rapid-acting treatment for depression in clinical studies (NCT03051256), it gained FDA Fast-Track designation as an adjunctive treatment for MDD. A phase III clinical trial of dextromethadone is currently ongoing (NCT04688164).

**Dimebon** (latrepirdine), a negative allosteric modulator at the polyamine-binding site of NMDARs, was originally used as an antihistamine [15]. The mechanisms of action of dimebon can also be explained by its effect on mitochondrial function and inhibition of cholinesterase [301]. In a phase III clinical trial, dimebon has shown negative results in mild to moderate Alzheimer’s Disease [302]. In a phase II trial in patients with HD, short-term administration of dimebon was beneficial for cognitive improvement (NCT00497159) [303], but a subsequent phase III trial with 403 patients was negative on all outcomes (NCT00920946; HORIZON) [304]. 

### 3.2. Glutamate and Glycine Site Antagonists

**Selfotel** (CGS-19755) is a competitive antagonist of the glutamate-binding site on NMDARs and is the most widely studied NMDAR antagonist for cerebral ischemia therapy, with an opportunity to widen the therapeutic window [305]. This molecule has successfully passed phases I and II clinical trials as a safe drug [306] but was, unfortunately, not an effective treatment for acute ischemic stroke, as determined in a subsequent phase III trial [307]. Indeed, a trend toward increased mortality, particularly within the first 30 days and for patients with severe stroke, suggests that the drug might have a neurotoxic effect on brain ischemia [307].

**Rapastinel** (GLYX-13) is an amidated tetrapeptide, partially agonist of glycine-binding sites on NMDARs [22]. It received a ”breakthrough therapy” designation from the FDA as an adjunctive treatment of treatment-resistant MDD for reducing depressive symptoms (NCT01014650) [22,308]. The developing company, Allergan, announced in 2019 that rapastinel failed to meet its endpoints in three acute MDD phase III clinical studies compared with placebo (NCT03668600). 

Aptinyx Inc. (Evanston, IL 60201 United States) has developed a platform of novel NMDAR modulators that are orally bioavailable and work via a novel mechanism: functional glycine-site partial agonist modulation of the NMDARs: **NYX-2925; NYX-783; NYX-458**. As an example, NYX-2925 may be capable of selectively increasing NMDAR activity in hypoactive regions by binding to all GluN2 subunits, preferentially GluN2B, and seems to have an inverted-U dose response in the Chronic Constriction Injury Rodent Model. NYX-2925 is a mimetic of rapastinel [23]. The modulator has been tested in a rodent model of neuropathic pain before being developed for the treatment of fibromyalgia (NCT03219320, NCT04147858) [309,310]. Regarding NYX-458, its administration resulted in rapid and long-lasting improvement in cognitive function across the domains of attention, working memory, and executive function in a primate PD model. The phase II study is currently recruiting to test the safety and tolerability of NYX-458 but also activity across multiple neurocognitive assessments in people with mild cognitive impairment and mild dementia associated with PD and dementia with Lewy bodies (NCT04148391) [311].

**Apimostinel** (GATE-202, NRX-1074) is another glycine-selective modulator of NMDARs [24], developed to be the rapastinel next-generation compound and sharing a similar mechanism of action. Benefiting from its molecular weight and oral stability, apimostinel is 100-fold more potent than rapastinel and is also well tolerated without psychotomimetic symptoms [312]. Apimostinel is administered intravenously and orally and is undergoing efficacy and safety evaluation for MDD patients and healthy individuals (NCT02067793 and NCT02366364). The findings of the studies are not available yet.

**Gavestinel**. Among glycine-binding site antagonists, gavestinel (GV 150526) was fully investigated for stroke therapy. In animal models, this component showed a significant reduction of infarct volume and was able to protect somatosensory evoked potential responses [25]. Two large phase III clinical trials were conducted on ischemic stroke. Both were neutral, showing expected tolerance but no efficacy [313].

**AV-101** (L-4-chlorokynurenine), a pro-drug of 7-chlorokynurenic acid (7-CKA), which is a glycine binding site antagonist, is able to cross the BBB. In preclinical studies, AV-101 demonstrated dose-dependent antidepressant-like effects in animal models [26]. However, AV-101 monotherapy failed to produce the anti-depressant effects in a phase II clinical study (NCT02484456). A comparative trial with 180 treatment-resistant depression subjects found that the AV-101 group did not significantly relieve overall depressive symptomatology. A larger phase II study has been completed, but the results are not available yet (NCT03078322). 

Because glutamate and glycine-binding site antagonists are non-selective NMDAR antagonists, it is difficult to avoid the adverse neuropsychiatric reactions caused by a high dose of these compounds. Glycine site antagonists failed to show neuroprotective efficacy in human clinical trials or even produced intolerable CNS adverse effects. The failure of these agents has been attributed to poor studies in animal models and to poorly designed clinical trials. NMDAR antagonism may also have hindered endogenous mechanisms of neuronal pro-survival and neurodegeneration. Many agents such as selfotel but also dimebon, rapastinel, and gavestinel were ineffective in clinical trials.

### 3.3. Positive and Negative Allosteric Modulators

**SAGE-718** is a derivative of the endogenous steroid 24(S)-hydroxycholesterol and a PAM of NMDARs developed by Sage Therapeutics Inc. (San Diego, CA 92123 United States). SAGE-718 activity induces LTP of synapses and thus is essential for learning and memory and has been tested to treat cognitive dysfunctions in patients with neurodegenerative diseases. SAGE-718 is proposed to induce significant improvements in cognitive performance in healthy volunteers who received ketamine, with no apparent side effects [3,31]. It has recently moved to clinical phase II for the treatment of HD, PD, and AD (NCT03787758, NCT04476017, NCT04602624).

**NP10679** (developed by NeurOp Inc.; Atlanta, GA 30303 United States)**.** Small changes in extracellular pH are observed during stroke, or TBI. Extracellular protons are potent allosteric inhibitors of NMDARs that can significantly impact GluN2B-containing NMDAR current amplitudes [19]. A set of new compounds has been designed to increase efficiency when the extracellular pH is decreasing, including **NP10679** [314,315]. The phase I study showed that NP10679 has a half-life of approximately 17 h with no serious adverse effects (NCT03565861; NCT04007263). In 2021, the U.S. The FDA has granted orphan drug designation to NP10679 for the treatment of subarachnoid hemorrhage.

**Neu2000**, also called nelonemdaz or salfaprodil, is derivated from sulfasalazine and developed by GNT Pharma (Giheung-gu, South Korea). It is part of the negative allosteric modulator of GluN2B-containing NMDAR and has been designed to prevent both NMDAR-mediated excitotoxicity and free radical toxicity [32,33]. A phase II clinical trial showed that Neu2000 could be safely administered before endovascular thrombectomy (EVT) and has a positive effect on tissue damage and clinical outcomes in patients with acute ischemic stroke [316] (NCT02831088). One of the important outcomes of this study is that new therapeutic options because multi-target neuroprotection might mitigate reperfusion injury in patients with acute ischemic stroke before EVT. Neu2000 is investigated in a phase III trial in acute ischemic stroke patients receiving endovascular treatment to remove clots within 12 h following stroke onset (NCT05041010).

### 3.4. Compounds Acting Downstream of NMDARs 

GluN2B-NMDARs are thought to be mainly located in the extrasynaptic compartment in the adult brain and underlie neuronal death when overactivated, such as during stroke, TBI, AD, or HD [20,95,274]. For these reasons, GluN2B subunits are thought to be linked to the detrimental activity of NMDARs.

The overactivation of NMDARs, followed by the potentiation of Ca^2+^ entry, leads to cascades of heterogenous signals generating secondary damage and cell death. The modulation of NMDAR downstream pathways is also based on the coupling with postsynaptic proteins, and their interactors constituted nanodomains. In this review, we mainly focused on postsynaptic density 95 kD protein (PSD-95) included in protein nanocomplexes at the synapse. PSD95 is a scaffold protein and central organizer of postsynaptic signaling complexes comprising glutamate receptors, ion channels, signaling enzymes, and adhesion proteins [317]. Based on the PDZ domains, PSD-95 assembles GluN2B and neuronal nitric oxide synthase (nNOS) into a macromolecular complex [317]. By interacting with PSD-95, GluN2B activates nNOS [37,64,318]. An extensive literature has shown that NO overproduction by nNOS leads to the death of neurons, with the first proof in 1999 [319,320].

Treatment strategies based on the modulation of downstream pathways of NMDARs are intrinsically difficult due to the need for the putative drugs to cross both the BBB and the cell membrane to reach their target. A technology has been used to diminish these limitations: the shuttle-mediated delivery of the therapeutic molecules by conjugation to compounds able to cross the membranes. These compounds, also called cell-penetrating peptides are short peptides (maximum 40 amino acids) of different chemical characteristics (frequently positive charged) that can get through cellular membranes mimicking endocytosis [321]. 

**Nerinetide**, also called NA-1 or Tat-NR2B9c, is a peptide formed by the Tat sequence followed by the 9 C-terminal residues of the NMDAR-GluN2B subunit containing the PDZ ligand. This compound is being developed by NoNo Inc. (Toronto, Ontario, M5V 1E7, Canada) for the treatment of ischemic stroke, TBI, and subarachnoid hemorrhage. A phase III clinical trial (ESCAPE NA-1) was completed in 2020. A neuroprotective effect seems to be demonstrated only in the patient cohort where NA-1 was not co-administered with Alteplase^®^ (a recombinant form of tPA). This can be explained as Alteplase^®^ cleaving NA-1, consequently drastically reducing its half-life [322,323] (NCT02930018). Nerinetide is currently undergoing another phase III trial to confirm the potential efficacy in patients not receiving thrombolysis (NCT04462536). It is worth mentioning that Tat-NR2B9c is safe and effective in the treatment of patients with iatrogenic stroke after endovascular aneurysm repair [35,324] (ENACT trial; NCT00728182). With uric acid, NA-1 is the only compound tested in the last five years in a randomized controlled trial in addition of the standard of care for stroke (thrombolysis and thrombectomy; NCT00860366) [34].

In the line of cell-penetrating molecules development, Bach and collaborators developed the N-dimer, which has a 1000-fold improved affinity for PDZ1-2 of PSD-95 compared with NR2B9c (i.e., NA-1) within *in vitro* experiments [325]. Compared with Tat-NR2B9c, Tat-N-dimer has been shown to reduce the infarct volume further and improve cognition upon administration to rodents in stroke preclinical models [36]. **AVLX-144**, a Tat-N-dimer developed by Avilex Pharma (another name for UCCB01-144), completed a phase I clinical trial in healthy volunteers (NCT04689035). Similarly, **ARG-007**, another polyarginine peptide with a similar mechanism of action to NA-1 and AVLX-144, is being developed by Argenica Therapeutics, with neuroprotective effects demonstrated in preclinical stroke models [326,327].

**ZL006** is reported to dissociate the GluN2B-PSD95-nNOS complex. This de novo small molecule was synthesized to selectively inhibit clot formation showing neuroprotective effects *in vitro* and reduced cerebral ischemic injury in mouse and rat stroke models. In addition, ZL006 is reported to cross the BBB and not affect the normal function of NMDARs and nNOS [38].

**Tramiprosate**, the active metabolite of ALZ-801 developed by Alzheon Inc. (Framingham, MA 01701 United States), is a natural small aminosulphonate compound obtained from various species of red marine algae. By inhibiting the interaction between PSD-95 and nNOS and preventing the translocation of nNOS from the cytosol to the membrane, tramiprosate reduces the infarct volume. The treatment time window of tramiprosate was at least six hours [328]. This compound has been evaluated in two phase III mild to moderate ADstudies and showed significant benefits to mild subgroups of APOE4/4 AD patients [39].

**Edaravone-dexborneol** is comprised of two active ingredients, edaravone and (+)-borneol, and has been developed as a novel neuroprotective agent with synergistic effects of both antioxidation that scavenges hydroxyl, peroxyl, and superoxide radicals and anti-inflammatory processes [329]. By interfering with ROS and reactive nitrogen species generation, edaravone reduces nitric oxide production, and the cytochrome c-mediated apoptosis involved in NMDAR-dependent excitotoxicity [40]. Edaravone alone has been recently approved by the FDA to treat patients with ALS [330]. It has been marketed in Japan since 2001 to treat acute ischemic patients within 24 h of a stroke attack and is widely used in this country and China. The combination of edaravone-dexborneol is currently in phase III of the clinical trial (TASTE-2; NCT05249920), where patients will be given edaravone-dexborneol concentrated solution for injections twice a day for 10–14 days [329]. This molecule is currently tested as a hemorrhagic stroke treatment (NCT04714177).

**TAT-CAPON and ZLc-002**. Carboxy-terminal PDZ ligand of nNOS (CAPON) is an adaptor protein of nNOS and activated by NMDAR signaling [41]. Tat-GESV (a fusion peptide comprising Tat and YAGQWGESV) [331], Tat-CAPON- 12C (a fusion peptide comprising Tat and the last C-terminal 12 amino acids of CAPON), and ZLc-002 (a small molecule inhibitor) are typical nNOS/CAPON uncoupling agents preventing the activation of nNOS toxic pathway [332]. Tat-GESV could inhibit ischemia-induced recruitment of CAPON to nNOS and decrease ischemic damage in a severe model of neonatal hypoxia-ischemia when injected intracerebroventricularly just after carotid occlusion [331]. Tat-CAPON-12C microinjected into the peri-infarct cortex 4 to 10 days after photothrombotic stroke significantly decreased the number of foot faults in the grid-walking task and forelimb asymmetry in the cylinder task [333]. Similarly, ZLc-002 systemically injected 4 to 10 days after tMCAO reversed the impairment of motor function. Interestingly, the therapeutic effect of Tat-CAPON- 12C and ZLc-002 in the delayed phase of stroke recovery is due to the regulation of neuroplasticity and not to direct neuroprotective effect [333].

**Bpv and TaT-K13**. Phosphatase and tensin homolog (PTEN) is an important tumor suppressor. A previous study demonstrated the involvement of PTEN in neuronal death processes by its direct interaction with GluN2B containing NMDARs and by inhibiting PI3K (phosphatidylinositol 3-kinase) signaling, known for its pro-survival effect [334]. PTEN also contains a PDZ-binding motif at its C terminus, and NMDAR activation triggers a PDZ-dependent association between PTEN and PSD-95. GluN2B, PSD-95 and PTEN may form a complex *in vivo* [335]. The overactivation of GluN2B-containing NMDARs induces the nuclear translocation of PTEN, a step leading to excitotoxicity, thus making PTEN a mediator of a neurotoxic cascade. This mechanism has been described in several neurodegenerative and neuroinflammatory disorders such as stroke, PD, MS, and AD [42,43]. A preclinical study indicated that intraperitoneal injection of the PTEN inhibitor bpv did not reduce infarction during the acute phases of ischemic stroke, but when administered daily for 14 days, starting at 24 h after tMCAO, long-term functional recovery of tMCAO mice was significantly improved [336]. Tat-K13, a short fusion peptide that flanks the K13 residue of PTEN, could interfere with PTEN nuclear translocation and systemic application of Tat-K13, even six hours after tMCAO, not only reducing ischemia-induced PTEN nuclear translocation but also strongly protected against ischemic brain damage [43].

### 3.5. Antibody Targeting tPA-NMDAR Interaction

**Glunomab**. Since 2001, the breakthrough hypothesis that tPA can promote NMDAR overactivation by a mechanism based on its interaction with the extracellular amino-terminal domain (ATD) of its GluN1 subunit has been debated and confirmed by different world-renowned academic teams [45,104,337,338,339]. Zhu and collaborators showed that GluN1 ATD is highly mobile and actively participates in defining the gating and pharmacological profile of NMDARs, suggesting that any ligand binding on GluN1 ATD may stabilize its opened or closed conformations [340]. Among these ATD-GluN1-NMDARs modulators, the extracellular serine protease tPA, expressed by neurons and other cell types such as vascular endothelial cells [109,116], has been reported to have pleiotropic functions through the CNS and vascular structure [112,114,341,342,343]. By interacting with GluN1, tPA promotes an over-activation of NMDARs inducing neuronal death dependently or independently of its proteolytic activity [45,338]. Accordingly, the use of tPA inhibitors protects neurons from excitotoxicity [344,345]. tPA can also selectively increase neuronal extrasynaptic NMDAR surface diffusion. This selective diffusion of NMDARs is the consequence of a direct interaction of tPA with a functionally critical single amino acid (lysine 178) within the GluN1 ATD. This interaction of tPA is the first and necessary step of a previously suggested two-step process, which subsequently also involves an arginine in position 260 [337]. By this mechanism, tPA promotes NMDAR-dependent Ca^2+^ influx and excitotoxic death, both *in vitro* and *in vivo* [45,46,101,102,103,338].

In 2007, a study evidenced tPA-mediated NMDAR modulation using, for the first time, an active immunization in mice to allow transient and specific prevention of tPA interaction with the GluN1 subunit of NMDARs. This immunization significantly reduced the severity of ischemic and excitotoxic insults in mouse brains. It demonstrated that *in vivo*, tPA controls NMDAR-mediated neurotoxicity and the encoding of novel spatial experiences by interacting with the GluN1 subunit [45,104,338]. These important results were repeated in 2010 with an active immunization strategy and in 2011 with proof of efficacy of a polyclonal antibody designed against GluN1 ATD in cerebral ischemic and hemorrhagic stroke models [101,217]. These studies were confirmed by using resonance plasmonic surface approaches, confirming the tPA/ATD-GluN1 interaction as a valid molecular target for a therapeutic strategy. Furthermore, a monoclonal antibody, Glunomab, which recognizes a fully conserved epitope on the ATD of the GluN1 subunit of NMDARs, has been designed to prevent the tPA/ATD-GluN1 interaction on both vascular endothelial and neural cells and is currently developed by the company Lys Therapeutics (14000 Caen and 69007 Lyon, France) for the treatment of neurovascular and neurodegenerative disorders. 

The activation of a subset of NMDARs expressed on brain endothelium has been revealed to contribute to harmful effects such as disruption of BBB permeability and the immune cell transmigration described in the physiopathology of MS and common with other CNS diseases [114]. Leucocyte transmigration after NMDAR activation is mediated by altered expression and distribution of tight junction proteins, which leads to disruption of the BBB [114]. By its direct interaction with the GluN1 ATD, tPA promotes signaling events, including the signaling pathways, which drive the increase in endothelial permeability, the subsequent immune cell transmigration, and cell death [46,111].

On vascular endothelial cells, this innovative antibody restores the BBB integrity preventing the Rho/ROCK signaling activation and blocking the transmigration of inflammatory cells and the associated neuroinflammation processes without modifying the T cell activation [107,112]. On neurons, Glunomab prevents the excitotoxic effects of tPA and also interferes with the diffusion of NMDARs outside the synapse by blocking the translocation of NMDARs from the synaptic to the extra-synaptic compartment, an important hallmark described in psychotic diseases such as SCZ and anti-NMDAR encephalitis [46,258]. In addition to its long half-life, a strong advantage of this strategy is that this antibody does not need to cross the BBB to exercise its full therapeutic effects on the CNS since its main molecular target is the constitutive GluN1 subunit present on vascular endothelial cells and regulating the BBB permeability to toxic molecules and immune cells. Moreover, in contrast to many drugs targeting the NMDARs, Glunomab is restoring the NMDAR signaling to its physiological level without causing any perturbation to its basal and required functioning.

## 4. Discussion

NMDARs are widespread in the CNS and are essential mediators of synaptic transmission and plasticity. They are also coupled to either cell survival or death signaling pathways implicated in neurodevelopment, neurotoxicity, and neurodegeneration. Well-known to be expressed in neurons, NMDARs are also found in glial and vascular endothelial cells, where their functions are less understood. It is possible that experiments performed *in vivo* using pharmacological approaches targeting NMDARs and having concluded that neuronal functions were, in fact, nonneuronal related events or mixed events also involving glial cells, endothelial cells, or immune cells, for instance. The aberrant NMDAR activity plays a pivotal role in regulating clinical symptoms. An important feature of NMDARs is its subtype diversity due to the subunits composition and cell type localization which results in the formation of receptors with different compound binding, sensibilization, speed of activation, and deactivation profiles. Some of these subtypes of NMDARs may display ionotropic signaling, as reported above, but also possible metabotropic signaling pathways [346]. 

NMDAR-induced excitotoxicity is a significant part of the underlying pathological mechanisms implicated in neurodegenerative diseases such as PD, AD, HD or ALS and also in neurological injuries such as stroke and TBI [95,96,97,98,137]. This explains the early efforts to design antagonists capable of preventing NMDAR hyper-function. However, NMDAR hypofunction appears to be also harmful. For example, as specified above, high doses of Ketamine in healthy volunteers can mimic the positive, negative, and cognitive symptoms of SCZ [297,298]. Furthermore, anti-NMDAR encephalitis, which is characterized by decreased NMDAR expression and function, induces psychosis, abnormal behavior, and cognitive impairment. NMDAR hypofunction has also been described in MDD patients [243]. Finally, loss-of-function mutations in genes encoding NMDAR subunits were found in patients suffering from a variety of neurodevelopmental disorders, including intellectual disability or autism [264,265].

The presence of NMDARs on endothelial cells close to the tight junctions brings scientists to ask about their roles in vascular dynamics and in the NVU [105,106,107,108]. Solid preclinical proofs revealed that endothelial NMDARs are likely important regulators of blood-brain and blood-spinal cord barriers maintenance and permeability along with oxidative stress, neurovascular inflammatory processes, immune cell transmigration, mitochondrial function, and NO generation. In this review, we have reported that all these pathological processes have been largely described in several CNS disorders [7,44,114]. A large number of clinical trials based on NMDAR antagonists or agonists for the treatment of neurological diseases have failed, either due to poor efficacy or severe side effects. These failures could be explained by the major role of NMDARs in brain functioning, the complexity of NMDAR regulation, and the fact that the drugs developed so far aiming to interfere with NMDARs were either not specific enough to subtypes or subfunctions of NMDARs, or too active, leading to complete inactivation of NMDARs instead of their modulation.

Another therapeutic strategy focuses on positive or negative allosteric modulators of NMDARs, yet with few clinical proofs of efficacy of this mechanism as the majority of these drugs are still at the preclinical stage, with efficacy results based on *in vitro* recombinant receptors models (e.g., electrophysiology experiments on oocytes over-expressing NMDARs). However, few are currently reaching clinical stages with promising preliminary results in neurovascular or neurodegenerative diseases (PAM: SAGE718 and NAM: Neu2000, Figure 2). 

Recently, the development of molecular drugs designed to interfere with the NMDAR/PSD-95/nNOS signaling pathway (such as NA-1 and its derivates AVLX-144 or ARG-007) has opened new avenues in this field of research. The mechanism of action of NA-1, AVLX-144, and ARG-007 is described as a prevention of the interaction of GluN2B-containing NMDARs with PSD95 and the activation of a deleterious signaling pathway. To date, the research literature described the GluN2B-NMDARs to be mainly present in the extrasynaptic compartment of mature neurons involved in the excitotoxicity neuronal death. At the same time, PSD-95 is principally contained at the postsynaptic level despite its contribution to the lateral diffusion of NMDARs. Nevertheless, NMDARs are not exclusively expressed on neurons and are also present, for example, on other neural cells or on vascular endothelial cells forming the BBB and where NMDARs regulate its permeability and the immune cell infiltration in the brain parenchyma. As a consequence, strategies targeting only the exclusively neuronal expressed PSD-95 occlude all non-neuronal NMDAR mediated dysfunctions, including the endothelial NMDAR regulating the BBB permeability and the associated neuroinflammation mechanisms.

Interestingly, a new strategy targeting both vascular endothelial and neural NMDARs with a monoclonal antibody, Glunomab, was developed to counteract the hyperactivation of NMDARs triggered by tPA to the ATD of the NMDAR-GluN1 subunit. Through this interaction, tPA, an important component of the pathophysiology of many neurovascular and neurodegenerative disorders [101,107,217], promotes NMDAR over-activation and subsequent toxicity on the BBB and neurons [46]. In inflammatory conditions, the presence of tPA potentiates vascular endothelial NMDAR activity close to the tight-junctions and the Rho/ROCK pathway leading to an increase in BBB permeability leading to the infiltration of immune cells into the brain [111,112,114,119]. Glunomab is counteracting both the neuroinflammation directly linked to the increase of the permeability of the BBB and the following transmigration of toxic inflammatory cells into the brain parenchyma and the associated excitotoxicity mechanisms on neuronal synapses. Importantly, Glunomab does not perturb the basal and physiological activity of NMDARs since targeting the blockage of its overactivation solely by both endogenous and exogenous tPA-related processes. With its well-characterized mechanism of action, this innovative strategy is currently being developed for the treatment of ischemic stroke and MS, with the potential to expand the spectrum of clinical investigations to other neurodegenerative, neurovascular and mental diseases. In 2022, its potential beneficial effects have been recently revealed in a mouse PD model [347].

In conclusion, a better knowledge of NMDAR functions, dysfunctions, and regulation may allow the development of more appropriate, specific, and well-tolerated NMDAR-targeted drugs for CNS disorders, with reduced side effects and higher therapeutic potential. Results of ongoing and future clinical trials with these later innovative strategies are highly waiting. 

## Figures and Tables

**Figure 1 ijms-23-10336-f001:**
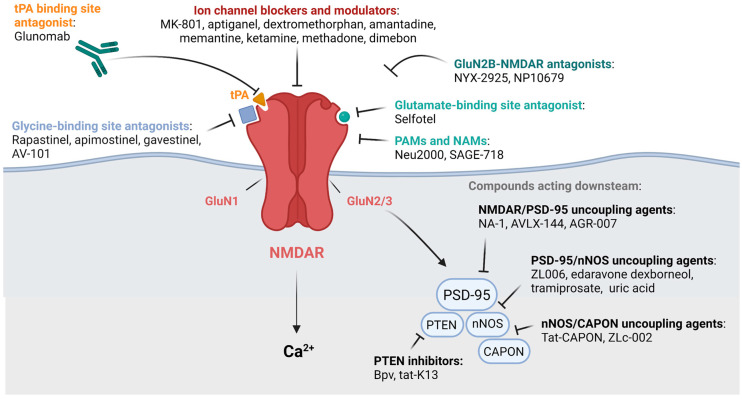
Pharmacological compounds targeting the NMDARs and their associated mechanisms. CAPON: Carboxy-terminal PDZ ligand of nNOS; PSD-95: postsynaptic density 95 kD protein; PTEN: phosphatase and tensin homolog.

**Figure 2 ijms-23-10336-f002:**
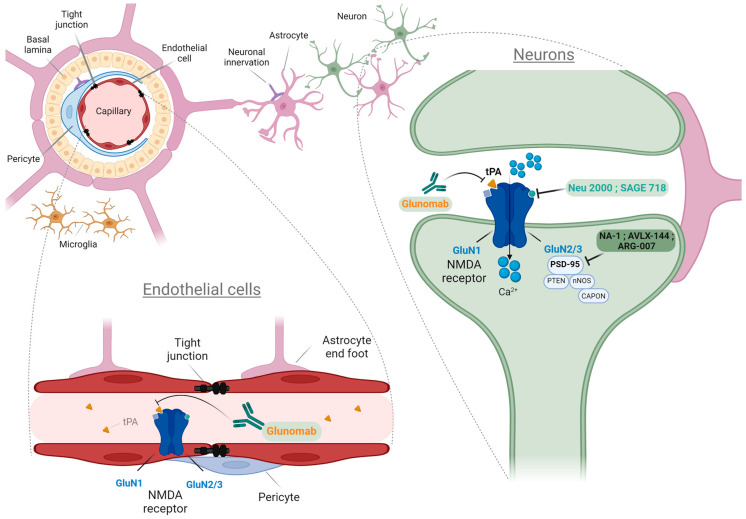
Potential mechanisms of action of innovative therapeutic drugs targeting either neuronal or endothelial NMDARs.

**Table 1 ijms-23-10336-t001:** Classification of endogenous and pharmacological molecules targeting NMDARs.

Endogenous Molecules	Pharmacological Molecules
**Channel blockers and modulators:** Mg^2+^ [1] Neurosteroids [8]	**Channel blockers and modulators:** MK-801 [9] Aptiganel [10] Dextromethorphan [11] Amantadine [12] Memantine [13] Ketamine [13] Methadone [14] Dimebon [15]
**Agonists:** Glutamate [16] Glycine/D-serine [17] L-Theanine [18]	
**Glycine/Glutamate site antagonists:** Polyamines [19] Zn^2+^; Cu^2+^ [20]	**Glycine/Glutamate site antagonists:** Selfotel [21] Rapastinel [22] NYX-2925 [23] Apimostinel [24] Gavestinel [25] AV-101 [26] Phencyclidine [27]
**PAMs and NAMs:** Spermines and Spermidines [19] Cholesterol [28] Pregnelonone sulfate [29] ATP [30]	**PAMs and NAMs:** SAGE-718 [31] NP10679 [19] Neu2000 [32,33]
**Downstream antagonist:** Uric acid [34]	**Downstream antagonists:** Nerinetide [35] AVLX-144 [36] ARG-007 [37] ZL006 [38] Tramiprosate [39] Edaravone Dexborneol [40] Tat-CAPON and ZLc-002 [41] Bpv and Tat-K13 [42,43]
**Modulators:** amyloid-β (Aβ) [44] tPA [45]	**Antibody:** Glunomab [46]

**Table 2 ijms-23-10336-t002:** NMDAR subtypes in resident and CNS infiltrating cells.

	Cell Types	Subunit Types
NVU	Neurons [20]	**GluN1**/GluN2(A,B,C,D)/GluN3 (A,B)
	Endothelial cells [70]	**GluN1**/GluN2A/2C/GluN3(A,B)
	Oligodendrocytes [71]	**GluN1**/GluN2C/GluN3(A,B)
	Astrocytes [72]	**GluN1**/GluN2(A,B,C,D)/GluN3(A,B)
	Microglia [73]	**GluN1**/GluN2(A,B,C,D)/GluN3A
	T cells [74]	**GluN1**/GluN2(A,B)
	Macrophages [75]	**GluN1**/GluN2(A,D)
	Neutrophils [76]	**GluN1**/GluN2B

## Data Availability

Not applicable.

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
