# Peer review of "Targeting NMDA Receptors at the Neurovascular Unit: Past and Future Treatments for Central Nervous System Diseases"

_ijms, 2022, doi:10.3390/ijms231810336_

Round 1

Reviewer 1 Report

The review article entitled 'Targeting NMDA receptors at the neurovascular unit: past and future treatments for central nervous system diseases' by Seillier and group, is very well written and covers the information related to the NMDARs function in CNS. Moreover, it covers the information related to the involvement of the sub-types of NMDARs in different neuropathological conditions like depression, schizophrenia, encephalitis, strokes, TBI, etc. The reviewer has two comments in minor to addressed before the paper can be accepted.

1. Although, authors have cited wide range of papers on NMDARs function, the important citation recently published by Wright et al., 2021 related to NMDAR antibody induced encephalitis is not been covered in this paper. Citation of this paper to review and discussing it briefly in the relevant section could cover beautiful piece of information on role of NMDAR-Ab in encephalitis. 

2. Authors have beautifully represented the pharmacological compounds targeting the NMDARs and their associated mechanisms. Adding the relevant information related the pathological conditions here in this diagram or adding one more representative diagram would help pulling the entire extract of review in one place.

Reviewer 2 Report

Seillier and colleagues present a comprehensive overview of NMDA receptor function within the context of central nervous system disorders. The authors have written an excellent account of NMDARs and their implications in disease, and in particular have provided a great insight into drugs that have been developed to target NMDARs, and how they might be utilised therapeutically. Generally, I find the review very well written. My suggestions for improvements centre on creating more parity between sections. For example, whilst the section on NMDARs in Stroke (2.2.1) contains lots of useful and well-cited information, other sections (e.g., 2.1.1 on Alzheimer’s disease ) is very brief and somewhat superficial. Overall this is a comprehensive review that will undoubtedly be of great use to the wider field.

 1. In the first section, the authors mention in passing metabotropic signalling mediated by NMDARs. The authors should include a citation here at the minimum, but they may also consider expanding this as this is relatively under-considered component of NMDAR function that could have important implications for our understanding and treatment of CNS diseases.

2. In Table 1 a diaeresis is included in ‘amyloid-b’, which would be considered atypical.

3. In section 1.1.3 the authors state that “Long-term potentiation (LTP) has been firstly described in 1983 by Collingridge and 114 collaborators [79].” I think what the authors mean here is that “The NMDAR-dependent form of long-term potentiation (LTP) was first described…”. LTP per se was described much earlier.

4. In general, the 1.1.3 could benefit from revising. Currently, it is a little vague and unclear (e.g., “LTD is defined as a reduction of the synaptic efficacy and a return to the basal 119 level, allowing a future information storage…”. A more general introduction to this section, outlining the role synaptic plasticity might play in learning and memory, the different forms of synaptic plasticity, and then how NMDARs have been implicated in the induction and expression of plasticity, would certainly help a reader less familiar with the topic. The authors might also consider differentiating between synaptic plasticity initiated by NMDARs (i.e., NMDAR-LTP), and synaptic plasticity of NMDARs (e.g., https://pubmed.ncbi.nlm.nih.gov/20852624/).

5. Section 1.1.4 describes the links between extrasynaptic NMDARs and survival/death signalling. The authors might consider discussing the inherent complexities to this relationship (https://pubmed.ncbi.nlm.nih.gov/23538441/) and that it is not simply the case of synaptic receptors are prosurvival and extrasynaptic are prodeath.

6. The section on Alzheimer’s disease (2.1.1) is interesting, but largely restates the commentary of Liu et al. (2019). Can the authors consider a more in-depth approach in synthesising the primary research as pertaining to the importance of NMDARs in AD?

7. The section on Parkinson’s disease (2.1.2) is quite brief and would benefit from a more indepth rationalisation of the involvement of NMDARs in the condition. Some points would benefit from further explanation (e.g., “Both PD-induced dopamine depletion and l-DOPA 321 treatment led to the redistribution of NMDAR subunits in levodopa-treated dyskinetic 322 rats and monkeys’ models.” Which subunits? Redistributed to where? To what effect?)

8. Section 2.1.4 on ALS is again interesting, but quite light on specific details linking ALS with NMDARs. The authors could consider explaining the specific details in their last sentence of that section “NMDARs seem [to] play an important role in ALS excitotoxicity based on their Ca2+ permeability and altered levels of their endogenous co-agonists/inhibitors”.

9. Line 485 – replace ‘gender’ for ‘sex’.

10. Section 2.4.1 on Depression – again, very brief. The only information really relevant to NMDARs is the last sentence. The authors could consider adding further explanation here.

11. Parts of the Discussion are restatements of preceding sections and could be made more concise or not included.

Reviewer 3 Report

The manuscript entitle, “Targeting NMDA receptors at the neurovascular unit: past and future treatments for central nervous system diseases” present an overview of NMDAR functions on cell types involved in the pathophysiology of neurodegenerative, neurovascular, mental, autoimmune and neurodevelopmental diseases. The authors have done a deep and comprehensive job. The review is well written, the table is well presented and it conclude with a nice discussion that summarizes the paper. Comments are as follow:

1-      Pericytes have recently attracted significant attention for regulating the development, functional integrity, and the maintenance of blood‐brain barrier. The author should include a mention regarding NMDARs on brain pericytes.

2-      Knowing that one of the most important features of NMDARs is subtype diversity and cell type localization, I recommend to include a table or figure that summarizes the NMDARs distribution in the different cells that form the Neurovascular Unit.

Author Response

Please see the attachment ( cover letter for the editor including responses to both reviewers)

Round 2

Reviewer 2 Report

The authors have addressed my suggestions.